# Apple Preload Halved the Postprandial Glycaemic Response of Rice Meal in Healthy Subjects

**DOI:** 10.3390/nu11122912

**Published:** 2019-12-02

**Authors:** Jiacan Lu, Wenqi Zhao, Linlin Wang, Zhihong Fan, Ruixin Zhu, Yixue Wu, Ying Zhou

**Affiliations:** Beijing Advanced Innovation Centre for Food Nutrition and Human Health, College of Food Science and nutritional Engineering, China Agricultural University, Beijing 100083, China; jiacanlu0613@163.com (J.L.); zhaowenqi@cau.edu.cn (W.Z.); lynn9523@126.com (L.W.); zhuruixin07@126.com (R.Z.); xiaoc0105@126.com (Y.W.); hpyzyzy@163.com (Y.Z.)

**Keywords:** glycemic response, preload, apple, satiety, rice meal

## Abstract

This study aimed to investigate the possible glycemic effect of apple preload on acute postprandial glycemic responses (GRs) of a following rice meal, comparing with its co-ingestion counterpart and an apple sugar solution preload, based on equal carbohydrates intake. In a randomized crossover trial, 18 healthy female subjects consumed (1) rice, (2) co-ingestion of apple and rice (A+R), (3) apple preload and rice (PA+R), and (4) rice with sugar solution preload (same sugar profile as in apple) (PSS+R). Acute postprandial GR tests and subjective satiety tests were carried out for each test food. Compared with rice reference, the PA+R achieved a 50% reduction of the iAUC_0-120_, a 51.4% reduction of the average peak value, and a 52.6% reduction of glycemic excursion in 240 min, while the PSS+R showed 29.7% and 31.6% reduction of peak value and glycemic excursion, respectively. No significant differences were found between R and PA+R in any of the satiety characteristics. Compared with rice control, apple preload of 15 g available carbohydrates remarkably lowered the acute postprandial GR without negative effect on satiety. The sugar component may partly contribute to the glycemic suppressing effect of the apple preload.

## 1. Introduction

Blood glucose management plays a role in the prevention of diabetes, cardiovascular diseases, obesity, certain cancers, and other diseases such as acne [1,2,3,4]. Among the postprandial glycemic management strategies, increasing attention has been on preload, as mounting evidence indicates that a small load of macro-nutrients ingested at a fixed interval before a meal can result in a milder postprandial glycemic and insulin response, higher postprandial satiety, and lower glycated hemoglobin (HbA1c), inflammatory markers, and serum lipids levels [5,6,7,8]. Although most of the preload studies focused on protein, there were limited studies that suggested that preloads of sugars may have the potential of mitigating postprandial glycemic responses (GRs) as well [9].

The World Health Organization report [10] and dietary guidelines of many countries called for a limitation of added sugar in daily diet [11,12,13] as the inclusion of added sugars in the diet would lead to a decrease of nutrition density, with fructose overconsumption being of particular concern [14,15,16]. However, fruit consumption is encouraged by health professionals as it is conducive to the prevention of cardiovascular diseases by providing antioxidants, potassium, and dietary fiber [17,18]. A number of studies indicate that the substitution of high glycemic index (GI) refined rice with certain fruits or dried fruits can improve postprandial GR [19,20], insulin responses [20], and HbA1c levels [21]. Some fruits rich in fructose such as apples have been proven to be low GI food [22], and the inclusion of fruit-source fructose to background diet based on isocaloric treatments reduced HbA1c in human trials [23]. A meta-analysis of observational studies concluded that temperate fruit consumption is negatively associated with the risk of diabetes and can be included in the diet of the diabetics [24].

A previous study reported that a kiwifruit preload resulted in a reduction of the incremental areas under the curve (iAUC) of postprandial GRs as well as curbed peak blood glucose values [20]. However, whether this glycemic effect exists when other fruits are used as preload is still unknown.

Apples are regarded as a low GI food [22] and a good source of polyphenols and pectin [25] in addition to fructose and glucose. According to epidemiological studies, the consumption of apple is beneficial to diabetes prevention [21,25,26]. As an annually available fruit, eating an apple prior to a meal is both convenient and well acceptable. However, to our knowledge, the glycemic and satiety effect of apple has not been put into study yet, either as a co-ingestion food nor as a preload food. Meanwhile, previous studies [27,28] suggested that the sugar profile seemed to be associated with glycemic and insulin responses, but the contribution of sugar to an attenuated postprandial GR is still to be confirmed.

This study aimed to investigate the possible glycemic effect of apple preload on the postprandial GRs of the following rice meal, compared with its co-ingestion counterpart and a sugar solution preload, based on equal carbohydrates intake. Our hypothesis is that (1) incorporation of apple into a rice meal by carbohydrate exchange is beneficial to postprandial GR compared to a single rice meal; (2) the apple preload treatment will better reduce postprandial GR of a high GI meal than its co-ingestion counterpart does; (3) the sugar component in apple contributes to apple’s glycemic effect; (4) the equal-carbohydrate partial substitution of apple for refined rice does not affect the satiety of the test meals.

## 2. Materials and Methods

### 2.1. Materials

All food materials were bought in the local supermarket. Red fuji apples (*Malus pumila* Mill.) were produced in Yantai, Shandong Province, China. White rice (*Oryza sativa* spp. *japonica*) used as reference food was cultivated and milled in Jilin, Jilin Province, China.

### 2.2. Subjects

Healthy young female volunteers aged between 18 and 25 were recruited through advertisements on the university bulletin board and BBS online. The questionnaires given to subjects contained the following criteria: (1) non-smoker; (2) non-drinker; (3) free from food allergies; (4) stable weight in the past three months; (5) regularly eating three meals and not breakfast skipper; (6) not on diet to gain or to lose weight; (7) not on medication in the past six months; (8) no metabolic disease or impaired glucose tolerance; (9) not in pregnancy. The sample size was calculated using the PASS 13 *Power Analysis and Sample Size* software (NCSS, Kaysville, UT, USA). It found that the test would have 85% power to examine a difference (*p* < 0.05) with 12 subjects in GI of 21.66, assuming that the standard deviation (SD) is lower than 8.41. These calculations were based on a two-treatment, randomized, crossover study where they observed a 21.66% reduction in GI for majia pomelo compared with the glucose control [29]. Twenty-four potential subjects who met these criteria were invited to the laboratory and involved in duplicated oral glucose tolerance tests (OGTT). The informed consent forms were signed by 18 eligible individuals. The study was conducted in accordance with the Declaration of Helsinki and carried out at the College of Food Science and Nutritional Engineering, China Agricultural University. The study protocol was approved by the Ethics Committee of China Agricultural University (ethics number CAUHR-2019002). Chinese Clinical Trial Registry registration number ChiCTR1900023948.

### 2.3. Study Design

The study applied a randomized crossover design where participants consumed test meals in a randomized order according to randomized numbers generated by computer on 14 separate mornings. Subjects were enrolled to this study using a single allocate ratio. The wash-out period was one week between each test session. Twenty-four hours before each trial day, the subjects were asked to refrain from drinking coffee or alcoholic beverages, excessive eating, staying up late, and strenuous exercise. Participants were instructed to consume the test meal in 10 min. Subjects were served 200 mL of water at room temperature at 120 min. They were provided with books, magazines, and Wi-Fi, and were suggested to stay seated during the test session and asked neither to consume food that was not related to the study nor to discuss the test meal. The glucose reference was tested twice respectively in order to assess the reliability of the study procedure.

### 2.4. Preparation of Test Meals

All test meals were given in portions containing 50 g of available carbohydrates: (1) glucose solution; (2) white rice with water; (3) co-ingestion of apple and rice (A+R); (4) preload apple, white rice consumed 30 min later (PA+R); (5) preload sugar solution (glucose and fructose accounted for 44.8% and 55.2% of the total sugar, respectively, sucrose was not detected, same as the sugar composition in apple) and white rice consumed 30 min later (PSS+R). In the A+R, PA+R, and PSS+R, apples or the sugar solution contributed 15.0 g of available carbohydrates and rice contributed 35.0 g. The glucose (50.0 g of Glucolin™ dissolved in 203 g of water in room temperature) and rice containing 50 g of available carbohydrates were prepared as dual reference foods. The composition of the test meals is shown in Table 1.

### 2.5. Blood Glucose Measurement

The glycaemic test protocol used was an adapted form recommended by the Food and Agricultural Organization (FAO) and the World Health Organization (WHO). The subjects were asked to arrive at the laboratory at 8:00 a.m., and their fasting plasma glucose concentrations were tested after a 10-min rest. The test meals were provided by a person who was not involved in data analysis to the subjects at 8:15 a.m., and the finger-prick blood samples were collected at 15, 30, 45, 60, 90, 120, 150, 180, 210, and 240 min following the start of the test meal. The second drop of blood was used for testing to avoid possible plasma dilution. Plasma blood glucose concentrations were measured on a ONETOUCH^®^ Ultra^®^ (LifeScan Inc., Milpitas, CA, USA) glucometer using the glucose oxidase method. The glucose oxidase method is considered to perform similarly to a standardized method in terms of evaluating blood glucose concentrations [30,31].

### 2.6. Measurements of Satiety

A visual analogue scale (VAS) [32] was used to assess the subjective satiety of the test meals during the test day. The VAS was 100 mm in length with words anchored at each end, expressing the most positive and the most negative rating. Subjects were asked to record their satiety when they finished the meals and at 15, 30, 45, 60, 90, 120, 150, 180, and 210 min after finishing the test meal (both preloads and meals) [33]. The use, reproducibility, and validity of the VAS have been described by Flint et al. [34]

### 2.7. Statistical Analysis

The GR data were converted to the value of glucose changes from the baseline. The incremental areas under the curve of postprandial GRs (iAUC), the incremental peak of blood glucose concentrations, the maximum amplitudes of glucose excursion in 240 min (MAGE_0–240_), and the negative area under the curve (NAUC) [35] were calculated. The iAUCs in different periods were calculated using the trapezoidal method [36], ignoring the area beneath the baseline level. The GI was determined from the iAUC areas of each test meal and that of the glucose control.

The satiety data were converted to the value of VAS changes from the baseline. The time of first negative rating (i.e., the time when a subject’s satiety fell below the fasting level) was recorded. The end time of eating was the average of each subject.

The statistical analysis was performed using the SPSS version 21.0 (SPSS Inc. Chicago, IL, USA) and the data were shown as the means (standard deviations, SD) or the means (standard errors, SE) where appropriate. The variables were assessed with the use of two-factor repeated-measures ANOVA, with treatment and time as factors. The test values between the meals were compared using a one-way analysis of variance ANOVA, and Duncan’s multiple range test was used to make adjustments for the multiple comparisons test. The criterion for statistical significance was a two-tailed *p* < 0.05. The correlation of data was determined using Pearson correlation analysis.

## 3. Results

### 3.1. Subject Enrolment

The study subject flow of blood glucose and satiety tests is shown in Figure 1. Eighteen subjects completed all the test sessions, and all their data were included.

### 3.2. Subject Characteristics

Subject baseline characteristics are shown in Table 2. The mean value of fasting blood glucose concentration was 5.3 (SD 0.2). In terms of fasting blood glucose values, there was no difference among all test foods and reference foods.

### 3.3. Blood Glucose

The GR for all test foods is shown in Figure 2. Compared with R, the A+R showed significantly lower incremental values at 30, 90, and 120 min, while PA+R elicited significantly lower glucose incremental values at 30, 45, 60, and 90 min. The PA+R further mitigated the postprandial GR with significantly lower incremental value compared with A+R at 30 min and 45 min. The glucose incremental values of PSS+R were significantly lower at 15 min, 30 min, and 45 min but higher at 210 min than that of R. The PSS+R had significantly higher glucose incremental values at 15 min, 60 min, and 120 min compared with PA+R. Both the PA+R and the PSS+R manifested small-double-peak style curves instead of one-sharp-peak curves in other test meals.

As shown in Table 3, in case of iso-carbohydrate partial substitution of apple for refined rice, the mixed meals elicited significantly lower GI values than rice meal did. The GI of PA+R was not only significantly lower than that of the rice reference but also lower than that of A+R and PSS+R. The iAUC of PA+R, A+R, and PSS was 50.5%, 22.5%, and 19.6% lower than that of rice reference. Compared with rice, the PA+R, A+R, and PSS+R resulted in a significant reduction in terms of incremental peak glucose value and MAGE_0-240_. The PA+R and PSS+R had a significantly greater reduction of peak value and MAGE_0-240_ compared with A+R. The incremental peak glucose value and MAGE_0-240_ of PA+R account for 51.4% and 52.6% of that of rice, respectively. There was no significant difference in negative area under the curve (NAUC), incremental low of glucose, peak time, and low time among the test meals.

### 3.4. Subjective Satiety

The values of VAS changed from the baseline are presented in Figure 3. A+R showed significantly higher incremental at 90 min compared with R, and at 90 min and 120 min compared with PA+R.

As shown in Table 4, A+R had higher iAUC_60-120_ compared with that of PA+R (*p* < 0.05). The iAUC_0-210_ of PA+R was 26.5% lower than that of A+R. Compared with rice and PA+R, A+R had a significantly higher incremental low of satiety. There were no significant differences between R and PA+R in any of the satiety characteristics.

## 4. Discussion

The present study has shown that adding apple to white rice meals, on the basis of equal carbohydrates intake, could suppress the acute postprandial GRs, and this effect was more pronounced when the apple was given as preload. Compared with the rice control, both the sugar solution preload and co-ingested A+R treatment significantly curbed the postprandial GRs in terms of iAUC_0-120_, the incremental peak of glucose and MAGE_0-240_. However, the apple preload containing only 15 g carbohydrate achieved a dramatic 50% reduction of the iAUC_0-120_, a 1.8 mmol/L reduction of the average peak value, and a 1.8 mmol/L reduction of glycaemic excursion in 240 min. The GI value of PA+R was only half of that of the rice reference.

It is not a surprise that the fructose-containing sugar solution could reduce the postprandial GR of the high GI meal when used in an equicarbohydrate exchange framework. Moore et al. reported that an acute fructose administration decreased the glycemic response to an oral glucose tolerance test in both normal and diabetic adults [37,38]. Heacock et al. reported that a sugar solution preload containing 10 g fructose cut the postprandial iAUC by 25% and 27% when consumed 30 and 60 min prior to a high GI potato meal [9].

A previous study observed a significant reduction in insulin response and an increase in glucagon level after a kiwifruit preload [20]. In a study on gastrointestinal hormone responses, it was found that the ingestion of 30 g fructose had little impact on insulin, GIP, and GLP-1 [39]. Inferred from the similar sugar profiles in reported kiwifruit (fructose:glucose:sucrose =2:2:1) and apple in the present study (fructose:glucose:sucrose = 6:5:0), the dramatic reduction of GR in apple preload treatment is not likely to be explained by insulin secretion pattern.

However, in the present study, the apple preload or sugar preload manifested small-double-peak style glucose curves instead of the conventional one-peak curve in the kiwifruit preload study. It is easy to understand that there was a small blood glucose rise after the ingestion of apple or sugar preload at 30 and 15 min, respectively. However, notably, the blood glucose level dipped in 45 min in both preload treatments, when it should have jumped up after rice consumption otherwise. This unique glycemic pattern suggested that the apple/sugar preload induced early insulin, which countered the tendency of blood glucose increase after rice ingestion. The underlined mechanism deserves further investigation and confirmation.

A meta-analysis of controlled intervention studies reported that the fructose-containing sugar from fresh fruit decreased HbA1c level without any negative metabolic effects on the basis of iso-carbohydrate substitution [23]. The beneficial effect of a small dose of fructose was explained by a ‘catalytic’ effect of fructose on hepatic glucose uptake [9], which resulted in decreased postprandial GR in high GI meals. A systematic review and meta-analysis of controlled feeding trials have shown that small doses (defined as <36 g/day based on three meals at <10 g/meal and two snacks at <3 g/snack) of fructose in exchange for other carbohydrates (mainly starch) decreased HbA1c by 0.4% [40].

In the present study, both the apple preload (143 g) or the sugar solution contained 6.7 g glucose and 8.3 g fructose, well within the scope of small doses. Since such a small dose of sugar made a difference to postprandial GR of a high GI diet, we inferred that the sugars in an apple are likely to act as an important factor in its GR effect. A previous study found that kiwifruit preload containing 25 g carbohydrate reduced the iAUC_0-120_ by 29.8%, cut the peak glucose value by 1.1 mmol/L, compared with the water-preload control [20]. However, the researchers concluded that it was the non-sugar components that improved GR profile to co-ingested cereal food, for the co-ingestion of kiwifruit sugar failed to reduce the GR of a wheat biscuit meal as the kiwifruit did [41]. A recent acute feeding trial failed to detect any positive effect on the glycemic response of oral glucose tolerance test (OGTT) in 25 healthy subjects when a 5 or 10 g fructose was added to the 75g glucose solution [42]. A study did not find any correlation between fructose moiety in dried fruits and the postprandial iAUC_0-120_ either when half of the rice meal was substituted by dried fruit containing 25g sugar [19].

The discrepancy between these results may partly be explained by the timing of fructose consumption. Comparing with the rapidly absorbed glucose, most fructose was absorbed slowly through the GLUT 5 by facilitated diffusion [43]. The great inter-subject variation reported in the OGTT test [42] might be due to the disparity of fructose absorption capacity, which differs considerably among individuals. The “catalytic” action of fructose on glucose metabolism might be better performed when the sugar was consumed 30 min prior to a high GI meal as shown by the preload study of Heacock et al. [9].

On the other hand, a small amount of glucose may help to mitigate the acute glycemic response as well. There is a report that preloads of sugar solution containing either 10 g or 20 g glucose at 30 min prior to meals lowered the blood glucose level after the meal [44]. The low dose glucose might elicit an early insulin response and raise the level of GIP [45], which would better prepare the body for a high GI carbohydrate meal. Hence, we surmise that a preload of the combination of glucose and fructose, such as the apple sugar solution used in the present study, may be more pronounced than either one of the two in terms of glycemic lowering capacity.

As mentioned above, after consuming the test meal, both the blood glucose curve of PA+R and PSS+R showed two small split-peaks which have not been found in the co-ingestion group. The identical pattern of glycemic curve observed in PA+R and PSS+R supports our hypothesis that the GR effect of apple preload can be partly explained by the sugar profile in it.

Compared with the sugar solution, the barrier of nature chewy structure could further delay the release of fruit-source sugar in the digestive tract. The non-sugar components in apple, including the insoluble fiber and pectin [46] and polyphenols [47], are possible factors contributed to apple’s slow digestion property, low GI value as well as the glycemic improving capacity. The difference of fiber content of rice and rice-apple meals (0.7 vs. 1.4) can partly explain the fact that in the co-ingestion A+R meal, the iAUC_0–120_ and peak value was significantly lower than that of the rice meal, though in a smaller extent comparing with its preload counterpart.

In this study, the weight of the test meals was fixed and female subjects with a similar appetite were used in order to minimize confounders in the satiety test. While there was no difference among treatments in terms of both the iAUC_0–60_ and iAUC_120–210_, comparing with the preload counterpart, the co-ingestion A+R treatment had higher iAUC_60–120_ and a higher incremental low of satiety. According to a previous study, the ingestion of fructose accelerated the gastric emptying rate [48]. In the co-ingestion treatment, the action of fructose on gastric emptying might be countered by a relatively large bulk of rice.

It is worth noticing that the satiety scores in the PA+R and PSS+R treatments were comparable to that of the rice reference, in spite of the disparity in protein content of the meals (6.2 g, 4.7 g, and 4.3 g in rice, PA+R, and PSS+R, respectively). The dietary fiber including pectin contained in apples might be one of the factors enhancing satiety [46] and slowing gastric emptying [49]. In addition, the subjective appetite is affected by eating speed and chewing [50], as chewing elicits the gut hormone response associated with satiety and reduces subjective hunger as well as food intake [51]. The increased satiety of A+R compared with rice reference might partly due to the need for more chewing when apple was added to the rice meal [51]. It can be expected that an apple preload will result in no negative consequences on satiety response in an iso-protein and iso-carbohydrate study design, while a co-ingestion treatment will lead to a higher level of satiety after a meal.

In this study, we achieved a halved postprandial glycemic excursion of a rice meal with an apple preload consumed 30 min prior to the meal. Such a remarkable GR reduction after a high GI meal by only a simple dietary approach was rarely reported. In order to investigate whether the sugar profile contributes to the glycemic effect of apple preload, we added the apple sugar solution preload treatment in the study. As far as we know, this is the first study to investigate the effect of a mixed glucose-fructose preload.

As a fruit, apple is particularly suitable to be used as preload food because it is well accepted, easy to carry, available all year round worldwide, and can be consumed either as a refreshing food between meals or as an ingredient of a salad appetizer. Substituting commonly used protein preload products with fruits such as apples will increase the intake of phytochemicals and micronutrients, which are beneficial to gut microbiota [52] and also aid in chronic disease prevention [53].

The present study was an acute trial conducted in healthy volunteers, so the result needs to be confirmed in pre-diabetes and diabetes patients. Another limit in the study design is that we did not include a hyper-carbohydrate treatment. An apple preload may increase the total carbohydrate intake if people do not reduce food intake in a meal after a preload. The insulin and gastrointestinal hormones, which would have provided more clues of the mechanism, were not investigated in the study. Further research is needed for a better understanding of the underlying mechanisms of the unique GR pattern of apple/apple sugar preloads.

In conclusion, the present study found that compared with rice control, apple preload of 15 g available carbohydrate remarkably lowered the acute postprandial GR without a negative effect on satiety. The sugar component can partly explain the glycemic suppressing effect of apple preload. Given the clinical importance of minimizing the glycemic excursion on decreasing the risk of type 2 diabetes and its associated complications [53], the metabolic potential of fruit preload prior to a high GI meal deserves to be tested in pre-diabetes and diabetic patients in the hope of developing a new easy and practical dietary approach to postprandial blood glucose management.

## Figures and Tables

**Figure 1 nutrients-11-02912-f001:**
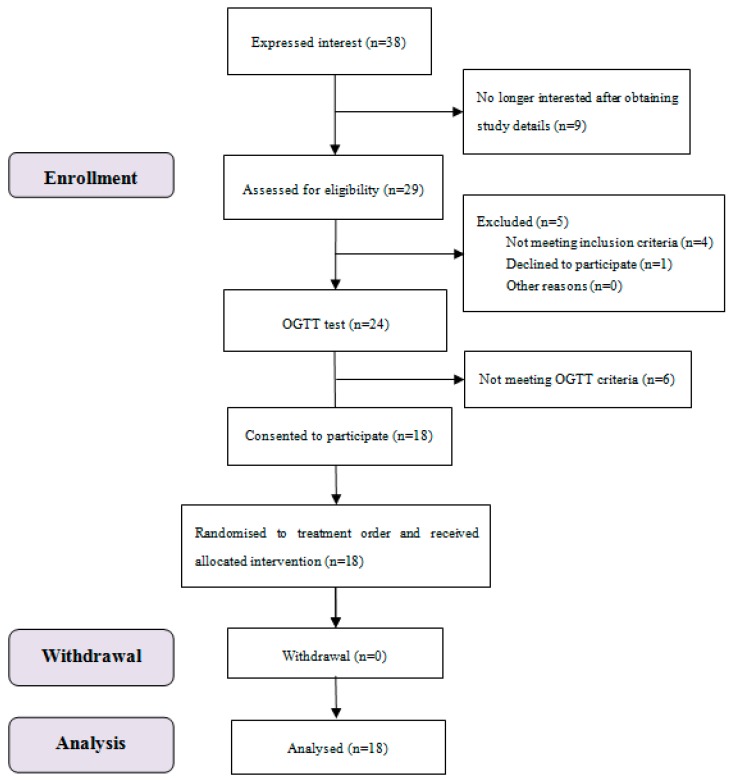
Consolidated standards of reporting trial (CONSORT) flow diagram of the study subjects.

**Figure 2 nutrients-11-02912-f002:**
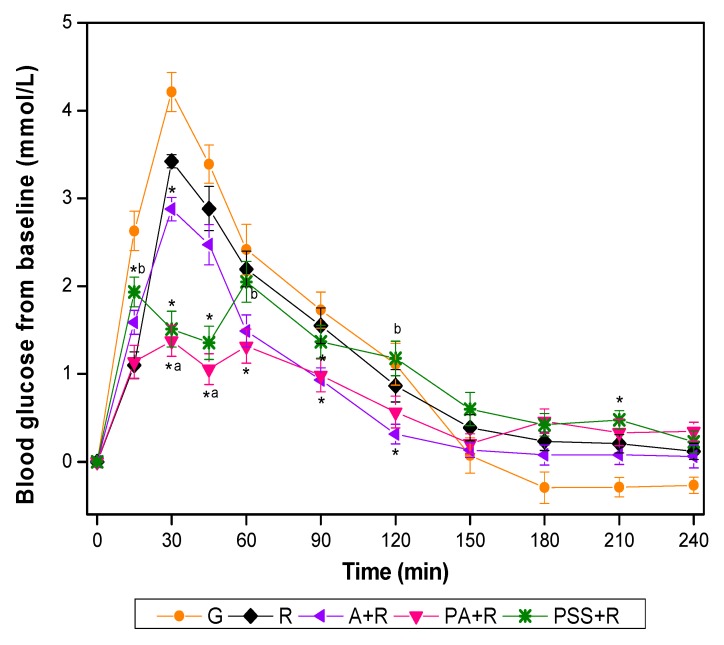
Blood glucose changes from baseline for test foods. G, glucose; A, apples; R, rice; A+R, co-ingestion of apples and rice; PA+R, preload apples; PSS+R, preload sugar solution. Values are the mean changes in blood glucose levels from baseline, *n* = 18, with their standard errors represented by vertical bars. * Test foods different from rice, ^a^ PA+R different from A+R, ^b^ PSS+R different from PA+R (*p* < 0.05).

**Figure 3 nutrients-11-02912-f003:**
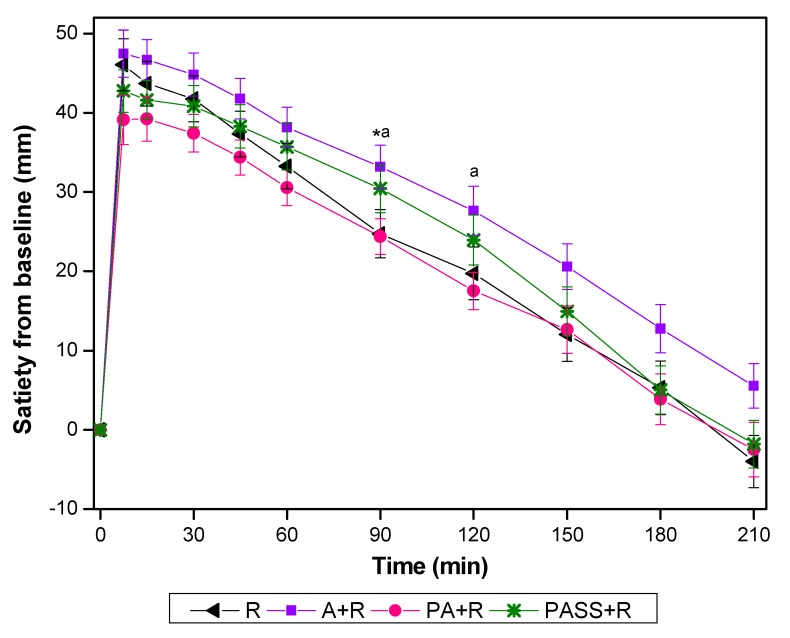
Satiety changes from baseline for test foods. A, apples; R, rice; A+R, co-ingestion of apples and rice; PA+R, preload apples; PSS+R, preload sugar solution. Values are the mean changes in blood glucose levels from baseline, *n* = 18, with their standard errors represented by vertical bars. * Test foods different from rice, ^a^ PA+R different from A+R (*p* < 0.05).

**Table 1 nutrients-11-02912-t001:** Composition of test meals (per serving) **^1^**.

Sample	Polished Rice (g)	Apple (g)	AC ^2^ (g)			Protein (g)	Fat (g)	Dietary Fiber (g)	Meal Size ^3^ (g)	Energy (kcal)
			Starch	Glucose	Fructose					
Glucose	-	-	-	50.0	-	-	-	-	258.5	200.0
Rice	176.0	-	50.0	-	-	6.2	0.5	0.7	258.5	228.8
A+R	115.7	142.8	35.0	6.7	8.3	4.7	0.9	1.4	258.5	227.0
PA+R	115.7	142.8	35.0	6.7	8.3	4.7	0.9	1.4	258.5	227.0
PSS+R	115.7	-	35.0	6.7	8.3	4.3	0.4	0.5	258.5	220.8

^1^ Nutritional data were obtained from manufacturers and determination experiments. ^2^ AC, available carbohydrate. Sucrose was not detected. ^3^ Including the water used for balancing the weight. A, apples; R, rice; SS, sugar solution; A+R, co-ingestion of apples and rice; PA+R, preload apples; PSS+R, preload apple sugar solution.

**Table 2 nutrients-11-02912-t002:** Subject baseline characteristics (Mean values with their standard deviations (SD) (n = 18).

Characteristics	Value		Reference Values
	Mean	SD	
Number of participants (n)	18	-	-
Age (year)	23.6	1.1	-
Body height (cm)	165.3	4.4	-
Body weight (kg)	54.2	4.9	-
BMI (kg/m^2^)	20.3	1.6	18.0~23.9
Fat mass (%)	25.7	3.0	17.0~30.0
Basal metabolism rate (BMR) (kcal/day)	1242.1	51.6	1210
Fasting blood glucose (mmol/L)	5.3	0.2	3.9~6.1

**Table 3 nutrients-11-02912-t003:** Postprandial glycemic characteristics of test meals in 240 min (Mean values with their standard errors (SE), n = 18).

Sample	iAUC_0–120_ (mmol/L·2 h)		iAUC_120–210_ (mmol/L·1.5 h)		GI		Incremental Peak of Glucose (mmol/L)		Incremental Low of Glucose (mmol/L)		Peak Time (min)		Low Time (min)		MAGE_0-240_ (mmol/L)	
	Mean	SE	Mean	SE	Mean	SE	Mean	SE	Mean	SE	Mean	SE	Mean	SE	Mean	SE
Glucose	276.7 ^a^	16.3	32.4 ^a^	6.9	100	−	4.4 ^a^	0.2	−0.8 ^a^	0.1	35 ^ab^	2	185 ^ab^	10	5.2 ^a^	0.2
Rice	220.0 ^b^	14.0	40.5 ^ab^	7.0	82 ^a^	6	3.7 ^b^	0.1	−0.1 ^b^	0.1	34 ^a^	2	193 ^a^	9	3.8 ^b^	0.1
A+R	170.6 ^c^	10.2	20.6 ^a^	6.2	64 ^b^	4	3.1 ^c^	0.2	−0.2 ^b^	0.1	35 ^ab^	2	193 ^a^	9	3.3 ^c^	0.1
PA+R	108.9 ^d^	13.0	39.1 ^ab^	9.5	40 ^c^	4	1.9 ^d^	0.1	−0.1 ^b^	0.1	41 ^ab^	5	127 ^c^	12	2.0 ^d^	0.1
PSS+R	176.9 ^c^	14.0	59.0 ^b^	9.7	69 ^ab^	7	2.6 ^e^	0.2	0.0 ^b^	0.1	46 ^b^	6	154 ^bc^	16	2.6 ^e^	0.2

The iAUC_0–120_ for postprandial GR was determined for the time interval in two hours using the trapezoidal method, ignoring the area beneath the fasting level; The iAUC_120–210_ for postprandial GR was determined for the time interval in one and a half hours using the trapezoidal method, ignoring the area beneath the fasting level; GI, glucose index; A, apples; R, rice; A+R, co-ingestion of apples and rice; PA+R, preload apples; PSS+R, preload sugar solution. Values are the mean glycemic characteristics of test foods, n = 18, with their standard errors. ^a,b,c,d,e^ Mean values within a column with unlike superscript letters are significantly different (*p* < 0.05).

**Table 4 nutrients-11-02912-t004:** Satiety characteristics of test meals in 210 min (Mean values with their standard errors (SE), n = 18).

Sample	iAUC_0–60_ (mm·1 h)		iAUC_60–120_ (mm·1 h)		iAUC_120–210_ (mm·1.5 h)		iAUC_0–210_ (mm·3.5 h)		Incremental Peak of Satiety (mm)		Incremental Low of Satiety (mm)		First Negative Time* (min)	
	Mean	SE	Mean	SE	Mean	SE	Mean	SE	Mean	SE	Mean	SE	Mean	SE
Rice	2176.7 ^a^	148.0	1485.1 ^ab^	164.2	945.8 ^a^	223.4	4607.6 ^ab^	497.0	45.8 ^a^	2.6	−3.8 ^a^	3.2	183 ^a^	11
A+R	2342.7 ^a^	134.7	1883.6 ^a^	154.2	1448.2 ^a^	239.3	5674.4 ^a^	495.9	46.9 ^a^	2.7	5.3 ^b^	2.7	200 ^a^	6
PA+R	1938.4 ^a^	124.0	1380.2 ^b^	118.8	851.4 ^a^	198.4	4169.9 ^b^	370.5	40.0 ^a^	2.7	−3.3 ^a^	2.6	180 ^a^	10
PASS+R	2073.7 ^a^	128.9	1671.2 ^ab^	162.4	1018.9 ^a^	204.4	4763.8 ^ab^	466.1	41.8 ^a^	2.3	−1.7 ^ab^	2.8	186 ^a^	7

* The first negative time was the time subjects’ satiety bellow the fasting level. The iAUC for satiety was determined for the time interval in corresponding time using the trapezoidal method, ignoring the area beneath the fasting level; A, apples; R, rice; ASS, apple sugar solution; A+R, co-ingestion of apples and rice; PA+R, preload apples; PASS+R, preload apple sugar solution. Values are the mean satiety feelings of test foods, *n* = 18, with their standard errors. ^a,b^ Mean values within a column with unlike superscript letters are significantly different (*p* < 0.05).

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
