# Peer review of "Apple Preload Halved the Postprandial Glycaemic Response of Rice Meal in Healthy Subjects"

_nutrients, 2019, doi:10.3390/nu11122912_

Round 1
Reviewer 1 Report
Major
The amount of rice consumed as a test meal R(ice) and in other test meals (A=R; PA+R; PSS+R) is significantly different: 176 g vs 115.7 g (starch: 50 g vs 35 g). It means that a part of rice (rice carbohydrates) was replaced by apple (apples carbohydrates). A beneficial effect was observed when a part of rice carbohydras was replaced by apple. There is also a significant difference in dietary fiber between R meal and A+R and PA+R meals (0.7 vs 1.4). It has to be clearly indicated in the Title and Abstract, taken into account during data interpretation, and clearly underlined under discussion and conclusion..
Minor
Compared to AUC the NAUC is not commonly used therefore some explanation is needed, and the difference in data interpretation between AUC and NAUC must be explained.
Fig.1 – no subject was excluded from the analysis – it must be indicated (=0) or only number of analyzed subjects should be presented
Line 158 – it should be SD not SE (as presented in table 2)
Fig.2 – the cure of changes in blood glucose after PSS+R is characterized by two peaks, it should be indicated and taken into account under discussion and the hypothesis of the mechanism of sugar effect (studied as “added” sugar preload) should be more deeply discussed.
It seems that mean and SE are presented, but it is not indicated in figure legend. This applies also to Fig.3. SE values are quite small. Could you refer to differences observed between study participants in their responses to different meals?
Reviewer 2 Report
This is a well written manuscript with and the authors conducted extensive analyses to present results from a randomized crossover trial demonstrating the differences in glycemic response and glycemic index measures following ingestion of apple coingestion with rice, apple pre-load with rice and sugar solution preload with rice each compared to rice alone meal.
Few minor comments:
In Table 2, add a column with population reference values of BMI, Fat mass, BMR and Fasting blood glucose. In LINE 166, the abbreviations for the exposures have been mis-entered. It should have been PA+R and PSS+R In Figures 2 and 3, it is difficult to distinguish between the chosen shapes to present the results in the plots, especially between darker and lighter forms of the same shape. Please choose different shapes or present a different type of plot In page 9 of 15 (incorrectly numbered as 2), the last paragraph has mis-entered abbreviations for exposures for PSS+R. Please correct those. In page 10 of 15 (incorrectly numbered as 3), the last line of the 5th paragraph can be written in a better way. (Suggestion: "which are beneficial to gut microbiota and also aid in chronic disease prevention.Author Response
Please see the attachment.
